# Epidemiology of disorders associated with tall stature in childhood: A 20-year birth cohort study

**Samuli Harju**[1,2]*, **Antti Saari**[1,2], **Reijo Sund**[1], **Ulla Sankilampi**[1,2]

**1** Institute of Clinical Medicine, School of Medicine, University of Eastern Finland, Kuopio, Finland,
**2** Department of Paediatrics, Kuopio University Hospital, Kuopio, Finland

\* samuli.harju@gmail.com

**Data availability statement:** The health data used in the study arise from the nationwide

## Abstract

Many primary and secondary disorders in childhood may cause tall stature (height of +2 standard deviations above the mean height for age and sex). Growth-monitoring programs are aimed at early detection of such disorders to avoid permanent health consequences and support children's wellbeing. However, age- and sex-specific data on the incidence of disorders associated with tall stature are scarce. This retrospective population-based cohort study aims to specify the epidemiological data that are needed to develop better diagnostic practices. The study population included 1 144 503 children (51% boys) born in Finland between 1998 and 2017 with 16.5 million register notifications including medical diagnoses. The first occurrences of several primary or secondary disorders associated with tall stature were identified from multiple registers. The age- and sex-specific cumulative incidences (CMIs) from birth until 18 years of age and the median age at diagnosis were determined. A total of 1641 children (0.14% of the whole birth cohort, 44% boys) had a primary or secondary disorder associated with tall stature. Klinefelter syndrome (47,XXY karyotype) was the most common primary disorder (median age at diagnosis: 8.4 years, CMI at 18 years: 1/2146 boys). Marfan syndrome (5.9 years, 1/4307 girls; 7.1 years, 1/5202 boys) and congenital overgrowth syndromes (1.7 years, 1/4717 girls; 1.8 years; 1/4925 boys) did not have a predilection for either sex. Secondary conditions such as central precocious puberty (1/894 girls at 8 years, and 1/4856 boys at 9 years) and hyperthyroidism (15.1 years, 1/936 girls; 14.4 years, 1/5675 boys) were more common among girls. Disorders associated with tall stature are rare and are frequently underdiagnosed in childhood. We suggest that during early childhood, the focus of growth screening should be particularly on Marfan syndrome and congenital overgrowth syndromes, with the addition of Klinefelter syndrome and central precocious puberty thereafter.

## Introduction

Tall stature is usually defined as a height more than +2 standard deviations (SDs) above the mean height for age and sex or height more than +2 SDs above the mid-parental target height (+1.6 SDs according to another definition) [1,2]. Excessive tallness in childhood is caused by

registers. The individual-level health data that support the findings of this study are neither publicly available nor to be shared because they contain potentially identifying and sensitive patient information. However, other researchers can request similar register data from the Finnish Social and Health Data Permit Authority (Findata; https://findata.fi/en).

**Funding:** The work was supported by: The Kuopio University Hospital State Research Funding (https://pshyvinvointialue.fi/web/en/state-research-funding), (SH, AS, and US); The Finnish Medical Foundation (https://laaketieteensaatio.fi/en/home/), (AS and SH); The Foundation for Paediatric Research (https://www.lastentautientutkimussaatio.fi/in-english/), (US and AS); and The Päivikki and Sakari Sohlberg Foundation (https://pss-saatio.fi/en/), (US, AS, and SH). The funders of the study had no role in the study design, data collection, analysis, interpretation, or writing of the manuscript.

**Competing interests:** The authors have declared that no competing interests exist.

several primary (congenital) and secondary (acquired) conditions, or if these are excluded, it is considered a normal variant of unknown cause (idiopathic) [3,4].

Primary growth disorders that often cause tall stature among other features include several clinically defined syndromes [3,4]. Examples of those include sex chromosome aneuploids, including 47,XXY or Klinefelter syndrome (KS), 47,XYY syndrome (XYY), and 47,XXX syndrome (triple X), which all lead to tall stature. They are estimated to be relatively common but highly underdiagnosed conditions with the maximum incidences of 1 per 500 in KS, 1 per 851 in XYY, and 1 per 1000 in triple X [5–9]. Growth acceleration in the prepubertal period is observed in Fragile X syndrome (FXS) [10,11]. Several monogenic syndromes, most importantly Marfan syndrome (MFS), are also associated with tall stature. Congenital overgrowth syndromes (COSs) such as Sotos and Beckwith–Wiedemann syndromes are rare and characterised by early overgrowth in addition to many other distinct features [12,13]. Accelerated growth is also frequently observed at the diagnosis of congenital adrenal hyperplasia (CAH), but it eventually leads to reduced adult height caused by accelerated bone maturation if left untreated [4,14].

Acquired (secondary) causes for pathological growth acceleration and tall stature in childhood comprise a few endocrinological conditions [3]. Growth hormone overproduction (pituitary gigantism) is a classical cause of tall stature, although it is extremely rare [15]. Common endocrine conditions such as hyperthyroidism are often associated with mild growth acceleration [4]. Central precocious puberty (CPP) is a frequent cause of growth acceleration and tall stature compared to healthy peers. It begins before age 8 in girls and age 9 in boys and requires prompt identification, investigations, and possibly management to prevent the early termination of linear growth and possibly short adult height [16].

Unlike short stature and its aetiology, which we have recently investigated in this same cohort, epidemiological data on conditions causing tall stature or growth acceleration are scarce [17,18]. Furthermore, previous studies have mostly investigated single disorders in smaller or restricted study populations. The present study aimed to elucidate the age- and sex-specific epidemiology of several primary and secondary disorders causing tall stature in a large population-based 20-year birth cohort. Such data are needed to develop evidence-based growth-screening programs and better diagnostic practices of disorders associated with tall stature in paediatric health care worldwide. Timely diagnosis of underlying conditions is important to prevent permanent effects on physical health and adult height and to support children's development.

## Materials and methods

### Study design, setting, and participants

This retrospective population-based 20-year birth cohort study examined 1 151 821 children born in Finland between 1998 and 2017 according to the Medical Birth Register (MBR), which is maintained by the Finnish Institute for Health and Welfare (THL) [19]. Over 92% of the children in the study population were of Finnish background [20]. The same study cohort and methods were used in our earlier study which investigated the epidemiology of disorders associated with short stature in children [17]. We combined data from four national health registers: MBR, the Care Register of Health Care (CRHC), the Statistics on Deaths (n = 4414), and the Purchases and Reimbursements for Prescription Medicines [21–23]. Subjects with incorrect personal ID codes or incomplete register notifications (n = 7318) were excluded. The final study population consisted of 1 144 503 children (51% boys) with 16.5 million care notifications including medical diagnoses. The first occurrences of the diagnoses for disorders associated with tall stature were identified from the MBR and CRHC. Data on linear growth measurements were not available.

The following primary or secondary disorders associated with tall stature were selected for analyses: KS (boys; ICD-10 codes Q98.0, Q98.1, Q98.4), XYY (boys; Q98.5), triple X (girls; Q97.0), FXS (Q99.2), MFS (Q87.4), COS (Beckwith–Wiedemann Q87.30, Sotos Q87.31, other Q87.38), CAH (E25.00, E25.01), hyperthyroidism (E05.0–E05.2), and CPP (E22.80 and E30.1). Idiopathic stall stature (ITS; familial or constitutional and non-familial: E34.40) was also included. There were only a few cases of pituitary gigantism (n = 4) and homocystinuria (n = 11) among the study population, which were not included.

## Statistical methods

The age- and sex-specific cumulative incidence (CMI, and 95% confidence interval) of the disorders was estimated from birth (diagnoses during the first week of life) until the maximum of 18 years of age, and the median age at diagnosis with interquartile range (IQR, from the 25th to 75th percentiles) was determined. Estimation was conducted using the cumulative incidence function (CIF) according to the Aalen-Johansen estimator, which estimates the cumulative probability of occurrence of an event of interest (the first diagnosis) in the presence of competing events [24]. Death before the end of the follow-up time was considered a competing risk, and follow-up was censored on the final day of 2017. We also checked the purchases of medications used regularly for KS (testosterone, ATC code G03BA03), CPP (leuprorelin, L02AE02), and hyperthyroidism (antithyroid preparations, H03B) to get a crude estimate of the number of children using the specific medication. Data were processed in SPSS version 27, and R statistical software version 4.3.2 (R Foundation) was used to calculate CIFs.

## Research ethics

Data were accessed for research purposes on 9 October 2018. After the initial exclusion process, data were pseudonymised, and thereafter only the research leader had access to information that could identify individual participants. There was no contact between the researchers and the study population. According to Finnish legislation, consent was not needed. The study was approved by the Research Ethics Committee of the Northern Savo Hospital District, and permissions to use data were obtained from the maintainers of the data (THL, Statistics Finland, and Social Insurance Institution).

## Results

This population-based study cohort examined 11.3 million person-years of follow-up in 1 144 503 children. In total, 1641 children (0.14% of the whole birth cohort, 44% boys) were diagnosed with either a primary (926 children, 65% boys) or secondary (715 children, 17% boys) disorder associated with tall stature before the age of 18 years.

### Primary growth disorders

Altogether 167 boys were diagnosed with KS, making it the most common primary tall-stature disorder among boys (Table 1, Fig 1). Of the diagnosed boys, 42 (25%) had at least one purchase of testosterone. 47,XYY syndrome (n = 96 boys) and triple X syndrome (n=68 girls) were more infrequent (Table 1, Fig 1). FXS was diagnosed in 172 children (77% boys) during the follow-up, and among boys, most diagnoses (80%) were made by the age of 6 years (Table 1, Fig 1).

The study population included 157 children (41% boys) diagnosed with MFS. The CMI of MFS was 23/100 000 in girls (95% CI 18.5–29.0) and 19/100 000 in boys (95% CI 14.1–26.0) at 18 years (Table 1, Fig 2). Congenital overgrowth syndrome (COS) was diagnosed in a total of 200 children (51% boys) during the follow-up (Table 1, Fig 2). Of the children with COS, 40.5% had Beckwith–Wiedemann syndrome, 40.5% had Sotos syndrome, and 19% had other

**Table 1. Cumulative incidences and median ages at diagnosis of primary and secondary disorders associated with tall stature and that of idiopathic tall stature.**

| Disorder (n) | Cumulative incidence per 100 000 children [95% CI] | | | | | | CMI[b] at 18 years | Median age at diagnosis, years (IQR[c]) |
|---|---|---|---|---|---|---|---|---|
| | Age[a], years | | | | | | | |
| | 0 | 2 | 6 | 10 | 14 | 18 | | |
| Klinefelter syndrome | | | | | | | | |
| Boys (167) | 6.2 [4.4, 8.5] | 13.2 [10.5, 16.5] | 18.8 [15.5, 22.8] | 27.2 [22.8, 32.3] | 34.0 [28.6, 40.3] | 46.6 [38.8, 55.7] | 1/2146 | 8.4 (1.1–14.4) |
| 47,XYY syndrome | | | | | | | | |
| Boys (96) | 2.6 [1.5, 4.2] | 4.3 [2.9, 6.3] | 11.0 [8.4, 14.3] | 17.9 [14.3, 22.3] | 23.1 [18.6, 28.6] | 26.1 [20.5, 32.9] | 1/3837 | 6.5 (3.8–12.0) |
| Triple X syndrome | | | | | | | | |
| Girls (68) | 3.2 [2.0, 5.1] | 6.4 [4.5, 8.8] | 10.3 [7.8, 13.5] | 13.0 [10.0, 16.6] | 15.1 [11.7, 19.4] | 15.9 [12.3, 20.6] | 1/6277 | 3.7 (0.1–8.2) |
| Fragile X syndrome | | | | | | | | |
| Girls (39) | 0.5 [0.2, 1.6] | 1.1 [0.5, 2.3] | 5.3 [3.6, 7.8] | 9.1 [6.6, 12.5] | 9.5 [6.9, 13.1] | 9.5 [6.9, 13.1] | 1/10500 | 5.7 (3.8–7.5) |
| Boys (133) | 0.2 [0.0, 1.1] | 6.4 [4.6, 8.9] | 22.8 [18.9, 27.3] | 25.8 [21.6, 30.7] | 28.3 [23.7, 33.7] | 28.3 [23.7, 33.7] | 1/3531 | 3.5 (2.2–5.0) |
| Marfan syndrome | | | | | | | | |
| Girls (92) | 0.5 [0.2, 1.6] | 7.4 [5.4, 10.0] | 12.0 [9.3, 15.3] | 17.2 [13.7, 21.5] | 21.2 [16.9, 26.3] | 23.2 [18.5, 29.0] | 1/4307 | 5.9 (1.3–10.0) |
| Boys (65) | 0.5 [0.2, 1.5] | 4.8 [3.2, 6.9] | 8.3 [6.1, 11.1] | 10.5 [7.9, 13.7] | 13.4 [10.1, 17.6] | 19.2 [14.1, 26.0] | 1/5202 | 7.1 (2.0–15.0) |
| Congenital over-growth syndrome[d] | | | | | | | | |
| Girls (99) | 0.7 [0.3, 1.8] | 10.9 [8.4, 14.0] | 17.5 [14.2, 21.5] | 19.3 [15.7, 23.6] | 21.2 [17.2, 26.0] | 21.2 [17.2, 26.0] | 1/4717 | 1.7 (0.7–5.3) |
| Boys (101) | 2.4 [1.4, 4.0] | 10.7 [8.3, 13.8] | 16.3 [13.2, 20.1] | 18.7 [15.3, 22.9] | 19.7 [16.0, 24.0] | 20.3 [16.5, 24.9] | 1/4925 | 1.8 (0.4–4.6) |
| Congenital adrenal hyperplasia | | | | | | | | |
| Girls (28) | 1.4 [0.7, 2.8] | 4.2 [2.7, 6.2] | 4.8 [3.2, 7.0] | 5.4 [3.7, 7.8] | 5.4 [3.7, 7.8] | 5.4 [3.7, 7.8] | 1/18474 | 0.1 (0.0–1.6) |
| Boys (40) | 0.7 [0.2, 1.8] | 3.4 [2.2, 5.3] | 6.2 [4.4, 8.7] | 7.3 [5.3, 10.1] | 8.1 [5.8, 11.1] | 8.7 [6.2, 12.1] | 1/11494 | 4.0 (0.1–7.3) |
| Hyperthyroidism | | | | | | | | |
| Girls (176) | 0 | 1.1 [0.5, 2.4] | 3.5 [2.2, 5.6] | 12.0 [8.9, 16.1] | 41.3 [33.8, 50.3] | 106.9 [90.1, 126.3] | 1/936 | 15.1 (12.7–16.6) |
| Boys (41) | 0 | 1.2 [0.6, 2.5] | 2.9 [1.7, 4,8] | 4.9 [3.2, 7.5] | 8.2 [5.6, 11.9] | 17.6 [12.0, 25.5] | 1/5675 | 14.4 (9.6–16.5) |
| Idiopathic tall stature | | | | | | | | |
| Girls (844) | 0 | 30.4 [26.0, 35.4] | 105.3 [96.3, 114.9] | 176.3 [163.7, 189.6] | 210.5 [195.9, 226.0] | 229.3 [213.2, 246.6] | 1/436 | 6.5 (3.6–9.6) |
| Boys (844) | 0 | 35.4 [30.7, 40.7] | 94.8 [86.6, 103.7] | 169.2 [157.2, 181.9] | 200.9 [187.0, 215.7] | 220.3 [204.5, 237.1] | 1/454 | 6.6 (3.5–9.6) |

[a]Age points 0, 2, 6, 10 and 14 years represent the ages at growth monitoring visits in the Finnish health care; age point 18 years gives the estimate of the final CMI of the conditions.

[b]CMI, cumulative incidence, expressed as 1/whole number

[c]IQR, interquartile range (from the 25th to 75th percentiles)

[d]Includes Sotos syndrome, Beckwith–Wiedemann syndrome, and other congenital overgrowth syndromes.

congenital syndromes including an early overgrowth phenotype (e.g., Weaver syndrome; see S1 Table for detailed information). Most diagnoses were made by the age of 6 years (81% in boys and 84% in girls).

CAH was diagnosed in 68 children (59% boys) during the follow-up, and most diagnoses were made by the age of 6 years (89% in girls and 72% in boys) (Table 1, Fig 2). Of the CAH diagnoses, 45 (66%) were the subtype E25.00 (salt-losing CAH) and 23 were subtype E25.01 (other or unspecified CAH, including 21-hydroxylase deficiency).

## Acquired growth disorders

The study population included 217 children (19% boys) diagnosed with hyperthyroidism by the age of 18 years (Table 1). Of these, 208 (95.9%) had thyrotoxicosis with diffuse goitre

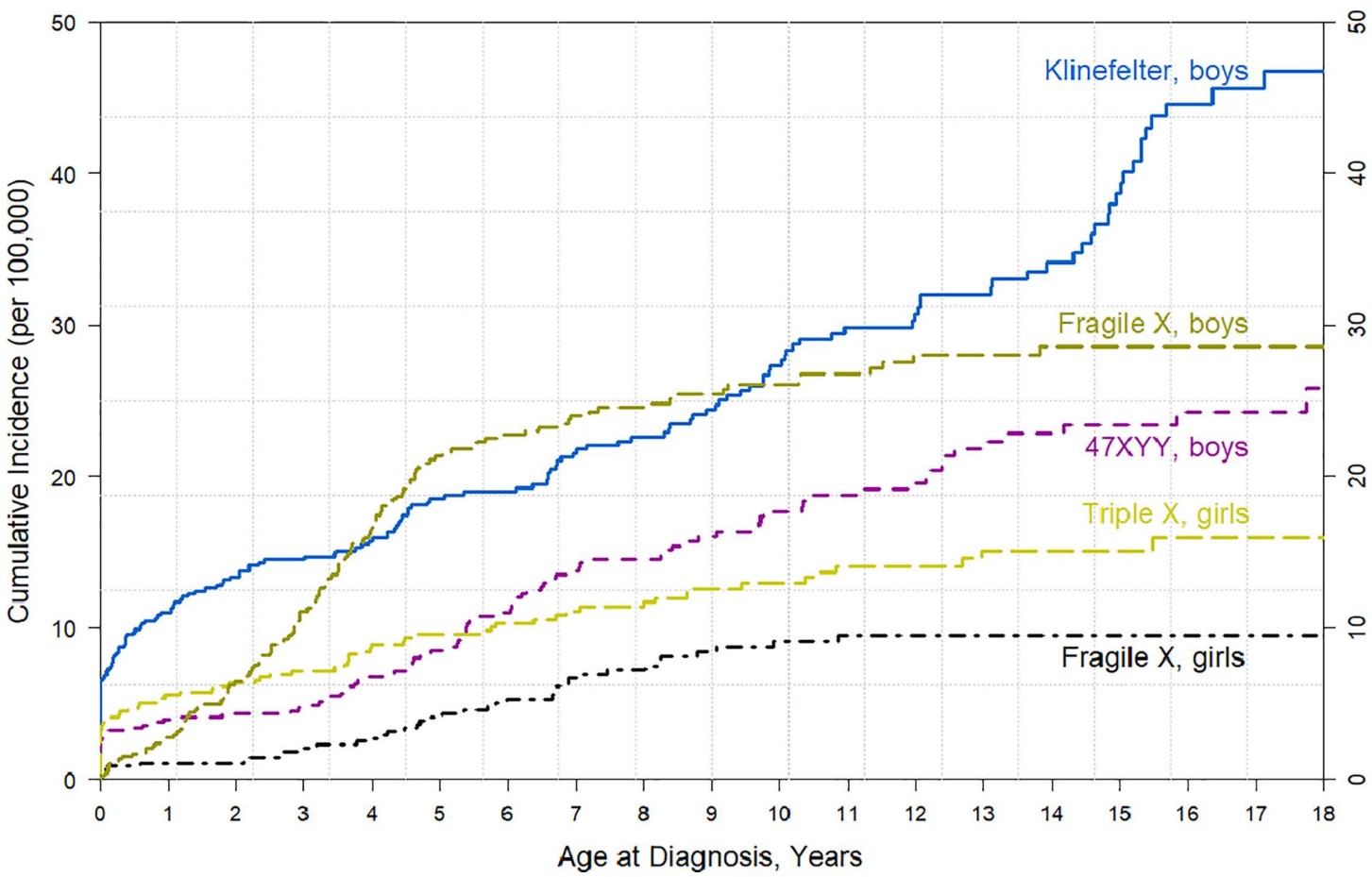

**Fig 1. Cumulative incidence of Klinefelter syndrome, 47,XYY syndrome, triple X syndrome, and Fragile X syndrome from birth to 18 years of age.**

(E05.0) and 9 (4.1%) had thyrotoxicosis with toxic nodular goitre (E05.1-2). Of the diagnosed children, 151 girls (86%) and 27 boys (66%) had at least one purchase of an antithyroid preparation (carbimazole; 177 children).

CPP before 8 years in girls and 9 years in boys was diagnosed in 498 children (15% boys) (Table 2). Of the diagnosed children, 162 girls (38%) and 25 boys (32%) had at least one purchase of leuprorelin. The median age at the first purchase was 7.3 years for girls (range 1.4–10.2 years) and 7.4 years for boys (range 1.0–8.8 years).

### Idiopathic tall stature

ITS was diagnosed in 1688 children (50% boys) during the follow-up and was the most common diagnosis in the present study (50.7% of all diagnoses). At 6 years of age, the CMI of ITS was already 105/100 000 in girls (95% CI 96.3–114.9) and 95/100 000 in boys (95% CI 86.6–103.7) (Table 1).

### Discussion

This 20-year birth cohort study is among the first to report broad, nationwide age- and sex-specific epidemiological data on primary and secondary disorders associated with tall stature in childhood. Tall stature is rare: only 0.14% of the whole birth cohort of over 1.1 million children had a primary or secondary disorder associated with tall stature. This was only

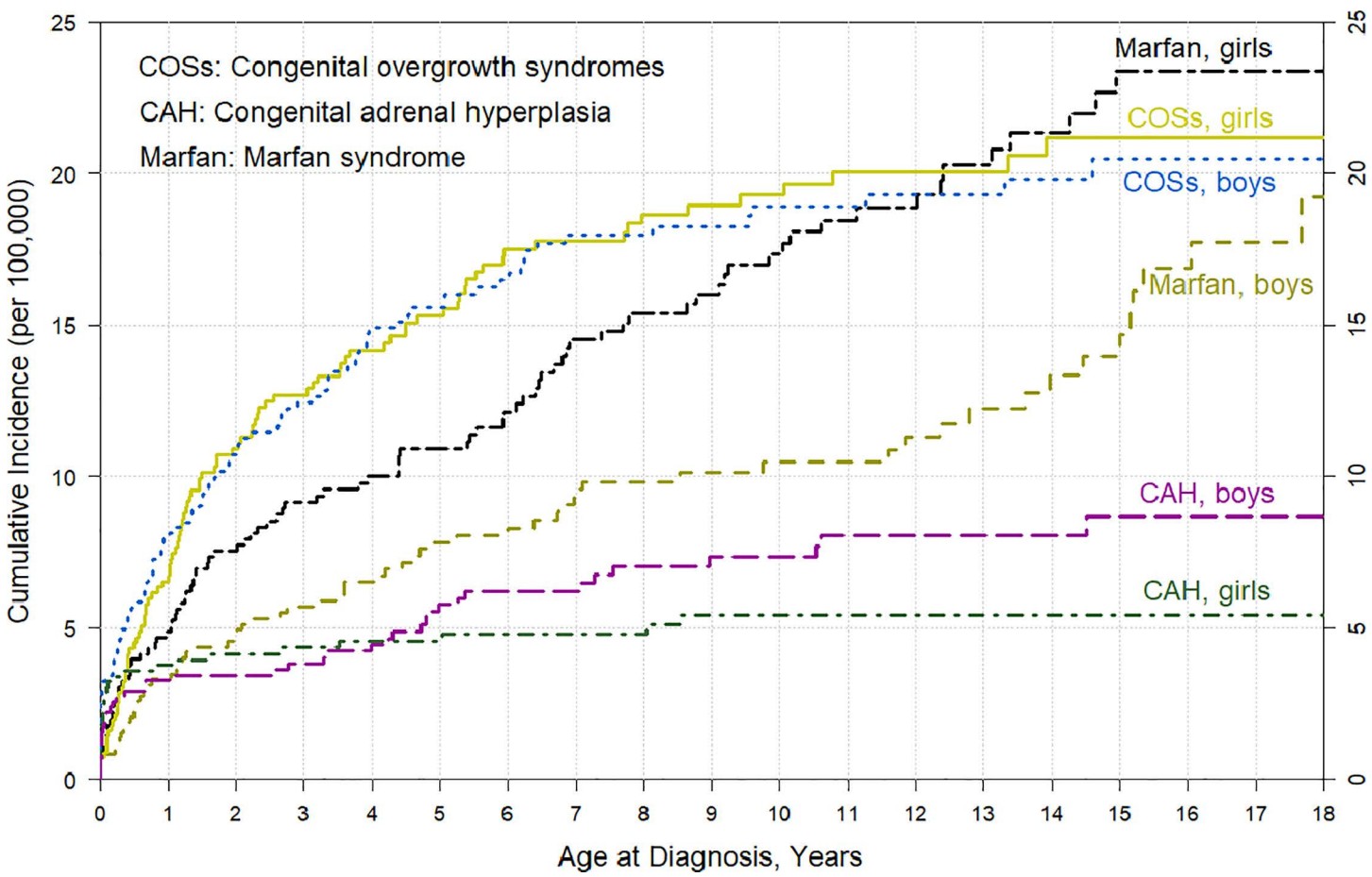

**Fig 2. Cumulative incidence of congenital overgrowth syndromes, Marfan syndrome, and congenital adrenal hyperplasia from birth to 18 years of age.**

Table 2. Cumulative incidence and median age at diagnosis of children diagnosed with central precocious puberty. By definition, puberty is precocious if observed before age 8 in girls and age 9 in boys.

| Disorder (n) | Cumulative incidence per 100 000 children [95% CI] | | | | | CMI at 8[a] or 9[b] years | Median age at diagnosis, years (IQR[c]) |
|---|---|---|---|---|---|---|---|
| | Age, years | | | | | | |
| | 0 | 2 | 6 | 8 | 9 | | |
| Central precocious puberty | | | | | | | |
| Girls (421) | 0 | 4.2 [2.7, 6.3] | 22.4 [18.4, 27.1] | 111.8 [101.5, 123.0] | | 1/894 | 7.3 (6.3–7.7) |
| Boys (77) | 0 | 0.5 [0.2, 1.6] | 3.8 [2.4, 6.0] | 10.4 [7.7, 14.1] | 20.6 [16.4, 25.7] | 1/4856 | 7.9 (6.6–8.6) |

[a]8 years among girls, and

[b]9 years among boys

[c]IQR, interquartile range (from the 25th to 75th percentiles)

one-sixth of the diagnoses associated with short stature found in the same birth cohort [17]. The results also show that tall stature disorders are probably underdiagnosed or the diagnoses are delayed, possibly reflecting wider social acceptance of tall stature than short stature

during childhood [25]. An early diagnosis is important for managing treatable comorbidities and for optimizing adult height. Thus far, there has been a lack of epidemiological data for the development of evidence-based growth-monitoring programs for tall stature. Based on these results, we suggest that in early childhood, the focus of growth screening should be particularly on Marfan syndrome and congenital overgrowth syndromes, with the addition of Klinefelter syndrome and central precocious puberty.

The general incidence of MFS is about 2–3 per 10 000 individuals and it was the most common primary cause of extremely tall stature in a Finnish cohort study which consisted of children born between 1990 and 2010 in Southern Finland [26,27]. In a large Taiwanese cohort study, Chiu et al. reported that the birth incidence of MFS is at least 1 per 4286 people [28]. Thus, the CMI of MFS obtained in the present study is closely comparable with that of other studies. Erkula et al. stated that in both sexes, a stature equal to the 95th centile (+2 SDs) of the general population was already passed at three years of age [29]. Early diagnosis and timely medical interventions are crucial because patients with MFS carry a high risk of morbidity and premature mortality. Consequently, MFS acts as an important target condition elucidating growth monitoring practices.

KS (47,XXY) is the most common sex chromosome aneuploidy and is estimated to occur in 100–200 per 100 000 males [8]. Underdiagnosis of KS is common and reflected also by the relatively low number of diagnoses in the present study: the CMI of KS was only 47 per 100 000 at 18 years. The median age at diagnosis was 8.4 years, and the diagnostic frequency was at its highest in pubertal years. A Danish study of pre- and postnatal prevalence of KS in children born between 1970 and 2000 reported that only about 10% were diagnosed before puberty [8]. Many cases are diagnosed later in adulthood when men are evaluated for infertility or hypogonadism, but up to three-fourths of cases remain undetected due to subtle phenotypes including tall and slender body structure [ 8,30]. Based on the present and previous observations, there is a need for improvement in the timely diagnosis of KS. Boys with KS often suffer from language and academic difficulties and psychological distress, which could be alleviated with proper support and neuropsychological care [31]. Furthermore, treatment with testosterone is beneficial for the normal development and maintenance of muscle and bone, as well as secondary sexual characteristics during and after puberty [32]. The present study indicates that the majority of boys diagnosed with KS do not receive testosterone medication.

Other sex chromosome aneuploidies associated with tall stature are rare. Two old studies from Denmark and the USA reported the incidence of XYY as varying from 1 per 851 to 1 per 3910 boys [9,33]. Thus, the CMI of XYY in the present study, 1 per 3837 boys at 18 years, is in the lower range. However, approximately 85% of males with XYY or more are never diagnosed, and most affected males have only a few phenotypic abnormalities, including tall stature [6,34].

The Danish study estimated the incidence of triple X to be as high as 1 per 1000 girls, which is six times more than the CMI obtained in the present study [9]. Although it is a congenital disorder, the majority remain undiagnosed [35]. Most triple X girls have only minor problems and subtle phenotype, but the final height is commonly at or above the 75th percentile [36,37].

Unlike chromosome aneuploidies, congenital overgrowth syndromes (COS) often have other distinct features in addition to tall stature, which is also a distinct feature in early life. In fact, our results showed that over 80% of COS cases were diagnosed before the age of 6 years. There is very little data on the epidemiology of COS, and there are no previous data on sex- and age-specific incidence values. We found that the CMI of COS at 18 years is at the same level as that of MFS, and there are no sex-specific differences.

In a large systematic review and meta-analysis published in 2014, FXS was estimated to occur in 1/7134 males and 1/11 111 females in total [38]. The present study offers CMI values from

birth until the age of 18 years when the CMI of FXS was 1/3531 for boys and 1/10 500 for girls. Notably, FXS is three times more common among boys, and 80% of FXS boys were already diagnosed by the age of six years, when the prepubertal growth acceleration may be seen, too [10].

Endocrine conditions associated with tall stature include CAH (congenital condition), and hyperthyroidism and precocious puberty (acquired conditions). Newborn screening for CAH was started in Finland only after the present birth cohort was collected. This is probably reflected in the higher median age at diagnosis in boys than in girls (4.0 versus 0.1 years) who often suffer from virilization of external genitalia at birth [39]. The CMI of 4.8/100 000 in girls and 6.2/100 000 in boys are in line with previous figures from the UK, indicating that approximately 1 of every 18 000 newborns has CAH [39]. A higher prevalence, 15.0 and 9.0 per 100,000 newborn females and males, respectively, was reported in a Danish study based on newborn screening, which also found milder forms of CAH [40]. A Finnish study showed that among a group of children with CAH diagnosed later in childhood, growth was already accelerated in infancy, but the adult height was low among boys [41]. However, newborn screening for CAH is now universal; therefore, CAH is not a primary target for growth screening.

Hyperthyroidism may cause acceleration in linear growth and advancement of skeletal maturation with the simultaneous loss of weight as an early sign of the condition [42]. Hyperthyroidism is rare in childhood but becomes more prevalent in adolescence as observed also in the present study. The rate of hyperthyroidism in children and adolescents in Northern Europe was 0.1 per 100 000 and 3 per 100 000 person-years, with a female-to-male predominance of 5:1 [43]. This is not directly comparable to CMI figures, which were considerably higher in our study. We observed that the difference between girls and boys in the CMI of hyperthyroidism already appears in childhood and becomes more evident over the years. At the age of 18 years, it was six times higher in girls compared to boys. The majority (86% of girls and 66% of boys) had at least one purchase of antithyroid medication, indicating distinct symptomatic disease and good diagnostic accuracy.

There are only a few studies on the prevalence and incidence of precocious puberty. A Danish register study of children born between 1993 and 2001 estimated that 0.2% of all girls (200 per 100 000) and less than 0.05% of boys (50 per 100 000) had some form of precocious pubertal development [44]. However, the results may be slightly overestimated because they included all forms of precocious puberty (CPP, premature thelarche, and premature adrenarche), and the age limit for CPP (9 years in girls and 10 years in boys) was extended by about 1 year. Another study examined 250 children (226 girls) diagnosed with CPP in Spain and revealed a low annual incidence of CPP ranging from 0.02 to 1.07 new cases per 100 000 [45].

The CMI values of CPP in the present study are between the values obtained in the previous two studies. Although incidence varies significantly among different populations, these studies show that precocious puberty is much more common among girls. Early development of secondary sexual characteristics and linear growth acceleration are important features of precocious puberty and should prompt further investigations [16]. The present study revealed that approximately one-third of children had purchased leuprorelin at the time of CPP diagnosis. Further studies are needed to assess their growth before diagnosis and the impact of medication on their final height.

Most individuals with tall stature are not diagnosed with any underlying pathological cause and are therefore diagnosed with ITS. In fact, ITS was the most common diagnosis in the present study (50.7% of all diagnoses). Girls and boys were diagnosed equally, which was also observed in another study [27]. Although ITS cannot be considered an actual target condition for growth screening, it is crucial to distinguish it from rare pathologic conditions. Consequently, unnecessary investigations could be avoided. However, ITS is a diagnosis of exclusion, and methods including molecular genetics might reveal pathology among children with ITS.

## Strengths

This study has several strengths. First, the study population was large with over 1.1 million individuals representing the whole child population born in 20 years. Second, the population was carefully monitored by one of the world's most extensive monitoring programs for childhood growth and health with over 20 check-ups [46,47]. If growth screening was abnormal, further examinations were performed systematically to diagnose the potential cause [48,49]. Therefore, it is probable that diagnoses of growth disorders are obtained in a timely manner in Finland. Third, the nationwide registers are shown to be good data sources [50].

## Limitations

Migration was not taken into account when calculating the CMI due to a lack of data. However, the number of emigrants/immigrants is known to be small, and disregarding migration is unlikely to influence the results significantly. We were not able to subdivide ITS into familial (or constitutional) and non-familial tall stature because the ICD-10 code for ITS does not differentiate between these forms, and fathers' heights were not available.

Furthermore, there can be differences in the genetic predisposition to various growth disorders between populations, so our data may not be generalizable to all populations. However, based on the discussed literature, this does not seem probable. Since we did not have access to the actual growth measurements, we can only speculate about how the yield of growth monitoring would mirror these data.

## Conclusions

This population-based study has provided the first age- and sex-specific epidemiological data on several primary and secondary disorders associated with tall stature. These disorders proved to be rather rare yet underdiagnosed in childhood. We suggest that during early childhood, the focus of growth screening should be particularly on MFS and congenital overgrowth syndromes, with the addition of KS and CPP thereafter. It is important to distinguish these pathological causes from ITS, which represented half of all cases associated with tall stature.

## Supporting information

**S1 Table. Congenital malformation syndromes involving early overgrowth: specific name and frequencies during the follow-up.**
(PDF)

## Author contributions

**Conceptualization:** Samuli Harju, Antti Saari, Reijo Sund, Ulla Sankilampi.

**Data curation:** Samuli Harju, Antti Saari, Ulla Sankilampi.

**Formal analysis:** Samuli Harju, Antti Saari, Reijo Sund, Ulla Sankilampi.

**Funding acquisition:** Samuli Harju, Antti Saari, Ulla Sankilampi.

**Investigation:** Samuli Harju, Antti Saari, Reijo Sund, Ulla Sankilampi.

**Methodology:** Samuli Harju, Antti Saari, Reijo Sund, Ulla Sankilampi.

**Project administration:** Antti Saari, Reijo Sund, Ulla Sankilampi.

**Resources:** Samuli Harju, Antti Saari, Ulla Sankilampi.

**Software:** Samuli Harju, Reijo Sund.

**Supervision:** Antti Saari, Reijo Sund, Ulla Sankilampi.

**Validation:** Samuli Harju, Antti Saari, Reijo Sund, Ulla Sankilampi.

**Visualization:** Samuli Harju, Antti Saari, Reijo Sund, Ulla Sankilampi.

**Writing – original draft:** Samuli Harju, Antti Saari, Reijo Sund, Ulla Sankilampi.

**Writing – review & editing:** Samuli Harju, Antti Saari, Reijo Sund, Ulla Sankilampi.

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
