## [Decision Letter · Decision Letter 0]

17 Oct 2024

PONE-D-24-32281Epidemiology of disorders associated with tall stature in childhood: a 20-year birth cohort studyPLOS ONE

Dear Dr. Harju,

Thank you for submitting your manuscript to PLOS ONE. After careful consideration, we feel that it has merit but does not fully meet PLOS ONE’s publication criteria as it currently stands. Therefore, we invite you to submit a revised version of the manuscript that addresses the points raised during the review process.

We look forward to receiving your revised manuscript.

Kind regards,

Bert B. Little, MA, PhD, FAAAS, FRAI, FRSM, FRSPH

Academic Editor

PLOS ONE

Journal Requirements:

2. We note that you have indicated that there are restrictions to data sharing for this study. PLOS only allows data to be available upon request if there are legal or ethical restrictions on sharing data publicly. For more information on unacceptable data access restrictions, please see http://journals.plos.org/plosone/s/data-availability#loc-unacceptable-data-access-restrictions. Before we proceed with your manuscript, please address the following prompts: a) If there are ethical or legal restrictions on sharing a de-identified data set, please explain them in detail (e.g., data contain potentially identifying or sensitive patient information, data are owned by a third-party organization, etc.) and who has imposed them (e.g., a Research Ethics Committee or Institutional Review Board, etc.). Please also provide contact information for a data access committee, ethics committee, or other institutional body to which data requests may be sent. b) If there are no restrictions, please upload the minimal anonymized data set necessary to replicate your study findings to a stable, public repository and provide us with the relevant URLs, DOIs, or accession numbers. For a list of recommended repositories, please see https://journals.plos.org/plosone/s/recommended-repositories. You also have the option of uploading the data as Supporting Information files, but we would recommend depositing data directly to a data repository if possible. We will update your Data Availability statement on your behalf to reflect the information you provide.

Additional Editor Comments:

Dear Authors-

Please revise according to the reviewers' comments.

In particular, the more specificity of the disorders that can be provided, the more value the manuscript will have for readers.

Please try to be as specific as possible, do no aggregate disorders. The reviewer is requesting disaggregation.

The revisions are minor, and we look forward to a revision in the near future.

Best regards,

Bert Little, PhD, FAAAS, FRAI, FRSM, FRSPH

Academic Editor

Reviewers' comments:

Reviewer's Responses to Questions

**Comments to the Author**

1. Is the manuscript technically sound, and do the data support the conclusions?

Reviewer #1: Yes

Reviewer #2: Yes

2. Has the statistical analysis been performed appropriately and rigorously? 

Reviewer #1: Yes

Reviewer #2: No

3. Have the authors made all data underlying the findings in their manuscript fully available?

Reviewer #1: No

Reviewer #2: No

4. Is the manuscript presented in an intelligible fashion and written in standard English?

Reviewer #1: Yes

Reviewer #2: Yes

5. Review Comments to the Author

Reviewer #1: The manuscript is a retrospective cohort study that investigated the cumulative incidence of primary and secondary disorders associated with tall stature in Finland over a 20-year period, from 1998 to 2017, among children up to 18 years of age. The findings suggest that some disorders remain undiagnosed. Special focus is given to Marfan syndrome, congenital overgrowth syndromes, Klinefelter syndrome, and central precocious puberty, which presented higher cumulative incidences.

The research appears to be well-conducted; however, the paper is somewhat confusing in regards to the diseases investigated. Specifically, there is inconsistency in the syndromes discussed across the abstract, introduction, and methods sections. Not all of the disorders are mentioned in each section, making it difficult for the reader to follow.

Minor issues:

The background section of the abstract could benefit from a clearer statement of the research aim.

In the introduction, adding more epidemiological information about the disorders investigated would be helpful, either individually or grouped as primary and secondary growth disorders if literature is scarce. Including global prevalence or incidence rates, especially in countries with similar characteristics to Finland, would strengthen this section. As it stands, it feels more like a list of syndrome definitions.

Make the importance of the research clearer and highlight why such rare conditions should be investigated.

In line 76, the term "essential disorders" could be clarified by listing the specific disorders the paper focuses on, helping the reader to know what to expect in the methods and results sections. Furthermore, in lines 93-99, not all of the investigated disorders are mentioned.

The discussion section is overly long and could benefit from summarization. The authors could prioritize key findings at the beginning, dedicating more space to discussing them in relation to existing literature. Less significant findings can follow, discussed in a more concise manner.

When comparing this manuscript's results with those from other studies, provide more context about those studies (e.g., location, study period, etc.), so that the reader can assess how comparable the results are.

Reviewer #2: Is the manuscript technically sound, and do the data support the conclusions?

I believe the manuscript is technically sound and the data support the conclusions because of the following:

- The problem, study aim, and possible benefits from this research is well defined and stated in the introduction section.

- Results support the suggestion that during early childhood, growth screening should take place among patients with MFS and congenital overgrowth syndromes, with the addition of KS and CPP in later years.

Has the statistical analysis been performed appropriately and rigorously?

- Aalen Johansen cumulative incidence function is appropriate. However, it is important explain why the presentation of age groups in table 1 and 2 is important. Please explain the rationality behind age categories and why it is different for table 1 and 2.

- I don't see the graphs of the functions in the manuscript although I see title for Figure 1 and 2. I believe it may be technical problem.

Overall, this is an interesting research. Please update the methods and results section acccordingly.

6. PLOS authors have the option to publish the peer review history of their article (what does this mean? ). If published, this will include your full peer review and any attached files.

**Do you want your identity to be public for this peer review?** For information about this choice, including consent withdrawal, please see our Privacy Policy .

Reviewer #1: **Yes: ** Caroline Zani Rodrigues

Reviewer #2: No

---

## [Author Response · Author response to Decision Letter 1]

29 Nov 2024

Response to Reviewers

Journal Requirements:

1. Please ensure that your manuscript meets PLOS ONE's style requirements, including those for

file naming.

AU: We have now checked that the manuscript meets PLOS One’s style requirements.

2. We note that you have indicated that there are restrictions to data sharing for this study. PLOS

only allows data to be available upon request if there are legal or ethical restrictions on sharing data

publicly. For more information on unacceptable data access restrictions, please see

http://journals.plos.org/plosone/s/data-availability#loc-unacceptable-data-access-restrictions. Before

we proceed with your manuscript, please address the following prompts: a) If there are ethical or

legal restrictions on sharing a de-identified data set, please explain them in detail (e.g., data contain

potentially identifying or sensitive patient information, data are owned by a third-party

organization, etc.) and who has imposed them (e.g., a Research Ethics Committee or Institutional

Review Board, etc.). Please also provide contact information for a data access committee, ethics

committee, or other institutional body to which data requests may be sent.

AU: Thank you for pointing this out. The permissions obtained from the maintainers of the

data (THL, Statistics Finland, and Social Insurance Institution), as required by the Finnish

legislation, do not allow the sharing of data. Some of the conditions associated with tall

stature are extremely rare and the data may contain potentially identifying or sensitive

patient material. However, other researchers can request similar register data from the

Finnish Social and Health Data Permit Authority (Findata; https://findata.fi/en).

We have now rewritten the data-sharing statement in the revised manuscript.

3. Please include captions for your Supporting Information files at the end of your manuscript, and

update any in-text citations to match accordingly.

AU: Caption for Supporting Information is now included at the end of the manuscript and

in-text citations match accordingly.

4. Please review your reference list to ensure that it is complete and correct. If you have cited papers

that have been retracted, please include the rationale for doing so in the manuscript text, or remove

these references and replace them with relevant current references. Any changes to the reference list

should be mentioned in the rebuttal letter that accompanies your revised manuscript. If you need to

cite a retracted article, indicate the article’s retracted status in the References list and also include a

citation and full reference for the retraction notice.

AU: We have now double-checked that we do not cite any papers that have been retracted

(until November 15, 2024). We have updated reference numbers 36 and 37 (in the former

reference list) to refer to the original studies. A few references became useless after the

revision and shortening of discussion and have been removed. The changes can be seen in

the “Manuscript with track changes”, which includes the former and current reference lists.

Additional Editor Comments:

Dear Authors-Please revise according to the reviewers' comments.

In particular, the more specificity of the disorders that can be provided, the more value the

manuscript will have for readers.

Please try to be as specific as possible, do no aggregate disorders. The reviewer is requesting

disaggregation.

AU: Thank you for these comments. We have now revised the manuscript based on the

reviewers’ comments. To be as specific as possible, we follow the ESPE (European

Society of Paediatric Endocrinology) classification of paediatric disorders associated with

tall stature (Wit JM, Ranke MB, Kelnar CJH. ESPE classification of paediatric endocrine

diagnoses. Horm Res. 2007;68(2):8-13). Each disorder is managed separately, and girls and

boys are presented as their own groups. Likewise in ESPE classification, congenital

overgrowth syndromes are handled as a single disease entity since they strongly overlap

each other and they are also very rare. However, information on the exact frequencies of

congenital overgrowth syndromes is given in the Supporting information at the end of the

manuscript. In some cases, disorders are grouped as primary (congenital), secondary

(acquired) or idiopathic tall stature. This division is important and it also has significance in

clinical work.

Reviewer #1:

The manuscript is a retrospective cohort study that investigated the cumulative

incidence of primary and secondary disorders associated with tall stature in Finland over a 20-year

period, from 1998 to 2017, among children up to 18 years of age. The findings suggest that some

disorders remain undiagnosed. Special focus is given to Marfan syndrome, congenital overgrowth

syndromes, Klinefelter syndrome, and central precocious puberty, which presented higher

cumulative incidences.

The research appears to be well-conducted; however, the paper is somewhat confusing in regards to

the diseases investigated. Specifically, there is inconsistency in the syndromes discussed across the

abstract, introduction, and methods sections. Not all of the disorders are mentioned in each section,

making it difficult for the reader to follow.

AU: Thank you for pointing this out. Throughout the manuscript we stick to the European

Society of Pediatric Endocrinology (ESPE) classification of tall stature disorders (Wit JM et

al, 2007) and to the basic division of these conditions into primary and secondary disorders.

Many of those conditions are rare. To improve the manuscript’s readability, we have now

limited the number of disorders presented in the Abstract to a few examples of the most

common primary and secondary conditions associated with tall stature. However, in the

other parts of the manuscript (Introduction, Methods and Results), more comprehensive and

detailed disease-specific data are presented to provide a thorough outline of various

conditions behind tall stature in childhood.

Minor issues:

The background section of the abstract could benefit from a clearer statement of the research aim.

AU: Thank you for the comment. We have revised the abstract so that the research aim is

more clearly stated: “This retrospective population-based cohort study aims to specify the

epidemiological data that are needed to develop better diagnostic practices.”

In the introduction, adding more epidemiological information about the disorders investigated

would be helpful, either individually or grouped as primary and secondary growth disorders if

literature is scarce. Including global prevalence or incidence rates, especially in countries with

similar characteristics to Finland, would strengthen this section. As it stands, it feels more like a list

of syndrome definitions.

AU: Thank you for the suggestion. We have now included general incidences of several

disorders (Marfan syndrome, Klinefelter syndrome, XYY syndrome, and triple X syndrome)

in the introduction. We also believe it is helpful for readers. However, it is important to

remember that those values are not directly comparable with the age- and sex-specific

cumulative incidences found in the present study. We did not find other studies reporting

age- and sex-specific cumulative incidences for the selected disorders.

Make the importance of the research clearer and highlight why such rare conditions should be

investigated.

AU: Thank you for your suggestion. We agree that many of the chosen conditions are rare

and it may raise questions regarding their value as research targets. We have now

added a short phrase to the abstract and we believe it makes the research aim and

importance clearer: ”This retrospective population-based birth cohort study aims to

specify the epidemiological data that is needed to develop better diagnostic practices.”

However, the epidemiology of disorders associated with tall stature is so poorly

characterized that it already gives us a basis to investigate it. Rare disorders may have

various lifelong consequences for the patient and his/her close relatives and should therefore

not be considered insignificant. Furthermore, this kind of long-term nationwide cohort study

is a reliable and often the only way to search the epidemiology of rare disorders. We try to

underline the importance of the study also at the end of the introduction: ”Such data are

needed to develop evidence-based growth-screening programs and better diagnostic

practices of disorders associated with tall stature in paediatric health care worldwide. Timely

diagnosis of underlying conditions is important to prevent permanent effects on physical

health and adult height and to support children’s development.”

In line 76, the term "essential disorders" could be clarified by listing the specific disorders the paper

focuses on, helping the reader to know what to expect in the methods and results sections.

Furthermore, in lines 93-99, not all of the investigated disorders are mentioned.

AU: Thank you for pointing this out. We have replaced the word ”essential disorders” with

”several primary and secondary disorders” that we have listed in the previous paragraphs.

We now list all the specific disorders focused on in the Methods (lines 96–99). Now

it reads as follows: ”The following primary or secondary disorders associated with tall

stature were selected for analyses: KS (boys; ICD-10 codes Q98.0, Q98.1, Q98.4), XYY

(boys; Q98.5), triple X (girls; Q97.0), FXS (Q99.2), MFS (Q87.4), COS (Beckwith–

Wiedemann Q87.30, Sotos Q87.31, other Q87.38), CAH (E25.00, E25.01), hyperthyroidism

(E05.0–E05.2), and CPP (E22.80 and E30.1). Idiopathic stall stature (ITS; familial or

constitutional and non-familial: E34.40) was also included.”

The discussion section is overly long and could benefit from summarization. The authors could

prioritize key findings at the beginning, dedicating more space to discussing them in relation to

existing literature. Less significant findings can follow, discussed in a more concise manner.

AU: Thank you for the suggestion. The discussion is now considerably shorter and rewritten

accordingly. We hope that it is now more readable and concise.

When comparing this manuscript's results with those from other studies, provide more context

about those studies (e.g., location, study period, etc.), so that the reader can assess how comparable

the results are.

AU: Thank you for your comment. We have now added more context to the referred studies

in the discussion section.

Reviewer #2:

Is the manuscript technically sound, and do the data support the conclusions?

I believe the manuscript is technically sound and the data support the conclusions because of the

following:

- The problem, study aim, and possible benefits from this research is well defined and stated in the

introduction section.

- Results support the suggestion that during early childhood, growth screening should take place

among patients with MFS and congenital overgrowth syndromes, with the addition of KS and CPP

in later years.

AU: Thank you for the positive feedback.

Has the statistical analysis been performed appropriately and rigorously?

- Aalen Johansen cumulative incidence function is appropriate. However, it is important explain

why the presentation of age groups in table 1 and 2 is important. Please explain the rationality

behind age categories and why it is different for table 1 and 2.

AU: In Table 1, the age groups 0, 2, 6, 10, 14, and 18 years are selected due to their clinical

representativeness based on the biology of growth and are also common growth check-up

points in Finnish health care. The regulation of linear growth in infancy (from 0 to 2 years)

switches to the childhood growth phase which is ongoing at 6 years (preschool age). The age

points 0, 2, and 6 years provide insight into the timely diagnosis of primary conditions.

Thereafter, at 10 and 14 years (the school health care checks), the acquired conditions are

increasing. The final age point, 18 years, gives us a rough estimate of the cumulative

incidence of the disorder near adulthood. We have now added information on the selection

of age points in the subtitle of Table 1.

Precocious puberty is presented separately in Table 2 for technical reasons because

the relevant age points differ from those used in Table 1. By definition, puberty is precocious

if it is observed before age 8 in girls and before age 9 in boys. Therefore, these age points

are used in Table 2. Data on the definition is now included in the title of Table 2.

- I don't see the graphs of the functions in the manuscript although I see title for Figure 1 and 2. I

believe it may be technical problem.

AU: We are sorry that you were unable to see the figures since we believe they are very

demonstrative when comparing different disorders and sexes. We had, however, uploaded

the figures in the required format. The version of the submission we received from the

submission system displayed the figures correctly, so we had no reason to believe that there

could be (technical) problems with them.

Overall, this is an interesting research. Please update the methods and results section acccordingly.

AU: Thank you. We have now updated the methods and result section as suggested

---

## [Decision Letter · Decision Letter 1]

2 Feb 2025

PONE-D-24-32281R1Epidemiology of disorders associated with tall stature in childhood: a 20-year birth cohort studyPLOS ONE

Dear Dr. Harju,

Thank you for submitting your manuscript to PLOS ONE. After careful consideration, we feel that it has merit but does not fully meet PLOS ONE’s publication criteria as it currently stands. Therefore, we invite you to submit a revised version of the manuscript that addresses the points raised during the review process.

Please note that I did not review the first version of this manuscript, and was invited to be the academic editor when the first AE was no longer available. Thank you for addressing the reviewers' comments. Please see below for my comments on the revised manuscript. ==============================

We look forward to receiving your revised manuscript.

Kind regards,

Heather Macdonald, Ph.D

Academic Editor

PLOS ONE

Journal Requirements:

Additional Editor Comments:

Please note that I did not review the first version of this manuscript and was invited to be the academic editor when the first AE was no longer available. The authors addressed the comments from both reviewers; however, I have some additional comments for consideration.

1. The authors mention in the statistical analysis section that they gathered information on medication purchase - what was the rationale for this, and why was this not included as a study objective? If it is just for descriptive purposes, please clarify in the text (earlier in the methods section - and include the same reference as in the 2022 paper if relevant). Also, results are provided for medications but this is not mentioned in the discussion.

2. Given the 20 year study period, did the authors investigate whether results differed by time - perhaps in 5-year increments or is this not relevant?

3. Regarding the national health registers, I don't have experience with these so am wondering if there is any instance where there would be discrepancies between registers? Or are individuals only in one register?

4. Is there a reference for the Aalen-Johnson estimator? I assume this estimator generates the 95% confidence intervals for the estimates? Please clarify this in the text.

5. Overall, I found that the text in the results section repeats the information in the Tables. Consider revising so the text highlights key findings without reiterating exact values that are presented in the tables. The 95% confidence intervals should be in the tables. I would also suggest including the interquartile range for the median age at diagnosis.

6. The references need reformatting to conform to PLOS One guidelines (e.g., in reference 4, "[Internet]" is not needed and neither is "Available from". The doi should be in the format: doi: 10.1136/archdischild-2013-304830 - please check formatting of all references.

7. The 2022 study of this same cohort should be mentioned earlier in the paper than the discussion. In particular, it should be cited in the methods section especially since the approach was similar between studies. I note that the 2022 paper included a hypothesis whereas this paper does not. In addition, the 2022 paper mentions that 92% of children in the study population were of Finnish background - this should be included in the current paper as well. Similarly, the 2022 paper notes that data on growth measurements were not available.

Reviewers' comments:

Reviewer's Responses to Questions

**Comments to the Author**

1. If the authors have adequately addressed your comments raised in a previous round of review and you feel that this manuscript is now acceptable for publication, you may indicate that here to bypass the “Comments to the Author” section, enter your conflict of interest statement in the “Confidential to Editor” section, and submit your "Accept" recommendation.

Reviewer #1: All comments have been addressed

Reviewer #2: All comments have been addressed

2. Is the manuscript technically sound, and do the data support the conclusions?

Reviewer #1: Yes

Reviewer #2: Yes

3. Has the statistical analysis been performed appropriately and rigorously? 

Reviewer #1: Yes

Reviewer #2: Yes

4. Have the authors made all data underlying the findings in their manuscript fully available?

Reviewer #1: Yes

Reviewer #2: Yes

5. Is the manuscript presented in an intelligible fashion and written in standard English?

Reviewer #1: Yes

Reviewer #2: Yes

6. Review Comments to the Author

Reviewer #1: (No Response)

Reviewer #2: The manuscript is in much better shape and ready for publication. Comments have been addressed as well.

7. PLOS authors have the option to publish the peer review history of their article (what does this mean? ). If published, this will include your full peer review and any attached files.

**Do you want your identity to be public for this peer review?** For information about this choice, including consent withdrawal, please see our Privacy Policy .

Reviewer #1: **Yes: ** Caroline Zani Rodrigues

Reviewer #2: **Yes: ** Shaminul Shakib

---

## [Author Response · Author response to Decision Letter 2]

10 Mar 2025

Additional Editor Comments:

1. The authors mention in the statistical analysis section that they gathered information on medication purchase - what was the rationale for this, and why was this not included as a study objective? If it is just for descriptive purposes, please clarify in the text (earlier in the methods section - and include the same reference as in the 2022 paper if relevant). Also, results are provided for medications but this is not mentioned in the discussion.

AU: Thank you for pointing this out. As you thought, the information on medication purchases is mostly for descriptive purposes, and thus, we did not consider it as our study objective.

We have now clarified the text in the method section as follows: “We also checked the purchases of medications used regularly for KS (testosterone, ATC code G03BA03), CPP (leuprorelin, L02AE02), and hyperthyroidism (antithyroid preparations, H03B) to get a crude estimate of the number of children using the specific medication.”

We also have some discussion on medication purchases (Manuscript lines 230–231, 272–274, and 287–290).

2. Given the 20 year study period, did the authors investigate whether results differed by time - perhaps in 5-year increments or is this not relevant?

AU: We thought about this during the early phase of the study. We also calculated incidence rates and incidence rate ratios of two different birth cohorts (e.g. birth year 1998–2007 versus birth year 2008–2017) with varying times of follow-up (e.g. from birth until the maximum of 4 years). The results revealed slightly more diagnoses of Marfan syndrome and congenital overgrowth syndromes in girls between different birth cohorts, but otherwise we did not find any significant differences. We decided not to include these results in the final paper. Although the study cohort of 1.14 million children is large, the number of diagnosed children is small due to the fact that these conditions are relatively rare. Further separation of boys and girls and comparison between different birth cohorts make the number of diagnosed children even smaller. Consequently, the cumulative incidence function from birth until the maximum of 18 years is statistically the most appropriate way to express the results.

On behalf of the Academic Editor, we received the message from the PLOS One Publications Assistant that we could disregard Comment #2. However, we decided not to remove the answer because we had already written it.

3. Regarding the national health registers, I don't have experience with these so am wondering if there is any instance where there would be discrepancies between registers? Or are individuals only in one register?

AU: Thank you for this interesting question. The national health registers used in the present study to search diagnoses were the Medical Birth Register (MBR) and the Care Register for Health Care (CRHC). The MBR forms the basis of the study because it includes all children born in Finland and their diagnoses during the first week of life. Since the exact time of diagnosis during the first week of life is not reported, we decided to set it as 0 years. After the first week of life, diagnostic data are recorded only in the CRHC. However, the CRHC also includes the diagnoses made during the first week of life, but there may be minor discrepancies in the time of diagnosis between the MBR and CRHC. This is mainly for technical reasons.

The CRHC also includes diagnoses of children who are not born in Finland. They were not included in the study because of the absence of diagnostic information at birth.

4. Is there a reference for the Aalen-Johnson estimator? I assume this estimator generates the 95% confidence intervals for the estimates? Please clarify this in the text.

AU: We now offer a reference concerning the Aalen-Johansen estimator (Coemans M, Verbeke G, Döhler B, Süsal C, Naesens M. Bias by censoring for competing events in survival analysis. BMJ. 2022 Sep 13;378:e071349. doi: 10.1136/bmj-2022-071349. PMID: 36100269).

Aalen-Johansen method estimates the cumulative incidence of the event of interest (accounting for competing events), and results are provided with the 95% confidence intervals. We have added 95% confidence intervals in Table 1 and Table 2.

5. Overall, I found that the text in the results section repeats the information in the Tables. Consider revising so the text highlights key findings without reiterating exact values that are presented in the tables. The 95% confidence intervals should be in the tables. I would also suggest including the interquartile range for the median age at diagnosis.

AU: Thank you for the comment. We have revised the results section accordingly. The 95% confidence intervals and interquartile ranges (from the 25th to 75th percentiles) have been added in Table 1 and Table 2.

6. The references need reformatting to conform to PLOS One guidelines (e.g., in reference 4, "[Internet]" is not needed and neither is "Available from". The doi should be in the format: doi: 10.1136/archdischild-2013-304830 - please check formatting of all references.

AU: References are now formatted according to PLOS One guidelines.

7. The 2022 study of this same cohort should be mentioned earlier in the paper than the discussion. In particular, it should be cited in the methods section especially since the approach was similar between studies. I note that the 2022 paper included a hypothesis whereas this paper does not. In addition, the 2022 paper mentions that 92% of children in the study population were of Finnish background - this should be included in the current paper as well. Similarly, the 2022 paper notes that data on growth measurements were not available.

AU: Thank you for the comments. We have made the revisions as suggested.

---

## [Editor Report · Decision Letter 2]

12 Mar 2025

Epidemiology of disorders associated with tall stature in childhood: a 20-year birth cohort study

PONE-D-24-32281R2

Dear Dr. Harju,

We’re pleased to inform you that your manuscript has been judged scientifically suitable for publication and will be formally accepted for publication once it meets all outstanding technical requirements.

Kind regards,

Heather Macdonald, Ph.D

Academic Editor

PLOS ONE

Additional Editor Comments (optional):

Thank you for addressing my comments.
---

## [Editor Report · Acceptance letter]

PONE-D-24-32281R2

PLOS ONE

Dear Dr. Harju,

I'm pleased to inform you that your manuscript has been deemed suitable for publication in PLOS ONE. Congratulations! Your manuscript is now being handed over to our production team.

Kind regards,

on behalf of

Dr. Heather Macdonald

Academic Editor

PLOS ONE